# Probiotics as Adjuvants in Vaccine Strategy: Is There More Room for Improvement?

**DOI:** 10.3390/vaccines9080811

**Published:** 2021-07-21

**Authors:** Diego Giampietro Peroni, Lorenzo Morelli

**Affiliations:** 1Department of Clinical and Experimental Medicine, Section of Pediatrics, University of Pisa, 56126 Pisa, Italy; 2Department for Sustainable Food Process–DiSTAS, Università Cattolica del Sacro Cuore, 29122 Piacenza, Italy; lorenzo.morelli@unicatt.it

**Keywords:** adjuvant, COVID-19, immune system, infant, LGG, microbiota, nutrition, probiotics, supplementation, vaccine

## Abstract

Background. It has been recognized that microbiota plays a key role in shaping immune system maturation and activity. Since probiotic administration influences the microbiota composition and acts as a biological response modifier, the efficacy of an adjuvant for boosting vaccine-specific immunity is investigated. Methods. A review of the literature was performed, starting from the mechanisms to laboratory and clinical evidence. Results. The mechanisms, and in vitro and animal models provide biological plausibility for microbiota use. Probiotics have been investigated as adjuvants in farm conditions and as models to understand their potential in human vaccinations with promising results. In human studies, although probiotics were effective in ameliorating seroconversion to vaccines for influenza, rotavirus and other micro-organisms, the results for clinical use are still controversial, especially in particular settings, such as during the last trimester of pregnancy. Conclusion. Although this topic remains controversial, the use of probiotics as adjuvant factors in vaccination represents a strategic key for different applications. The available data are deeply influenced by heterogeneity among studies in terms of strains, timing and duration of administration, and patients. Although these do not allow us to draw definitive conclusions, probiotics as adjuvants in vaccination should be considered in future studies, especially in the elderly and in children, where vaccine effectiveness and duration of immunization really matter.

## 1. Microbiome, Probiotics, Immune System and Vaccines

In recent years, much attention has been paid to the role of microbiota in shaping immune system development due to their impact on immune cell signaling and differentiation, since they are involved in driving both innate and acquired immunity [1]. This role has also been demonstrated in the modulation of immune responses to vaccination [2]. The microbiome can influence the microbial composition itself and the vaccine responses by different mechanisms: bacterial DNA, lipopolysaccharide (LPS) and flagellin can activate the Toll-Like-Receptor (TLR) pathways and, in turn, drive the dendritic cell (DC) responses. In mice, it has been shown that bacterial flagellin stimulated TLR5-mediated sensing that was necessary for antibody production to the influenza vaccine. Of note, TLR5-deficient mice, germ-free or antibiotic-treated animals had impaired responses to the influenza vaccine [3,4]. Furthermore, a deficit or disruption in microbial communities, such as in antibiotic-induced dysbiosis, is able to reduce the vaccine responses particularly in early life. This was demonstrated in animal models, where dysbiosis reduced specific IgG responses to different vaccines; this effect was reversed by the fecal transfer of commensal microbiota [5]. Notably, antibiotic administration did not alter the vaccine responses in adult mice, leading to the consideration that the infant microbiome is still developing and therefore characterized by low diversity, rapid change and high susceptibility [3].

Since probiotic administration can influence the microbiota composition and diversity, the use of probiotics as supplements has been proposed for immunological outcomes, such as clinical improvements of the immune responses [6]. It has been demonstrated that probiotics ensure, in a strain-specific way, immune modulation in viral infections with implications that are useful also when vaccine are considered. The concept that probiotics affect the vaccine response through modifications in the composition of the intestinal microbiota is supported by studies that verified changes in the stool microbiome after the intake of probiotics. Four small studies reported that stool microbiome changes, an abundance of particular strains and more bacterial diversity, induced by probiotic administration, were associated with greater responses to different vaccines [7,8,9,10]. 

Understanding the mechanism behind probiotic action provides a rationale for the selection for probiotics [11] (Figure 1). At the gut level, probiotics are able to influence the gut barrier function by effector molecules such as protein HM0539 by LGG that promotes the expression of tight junction proteins [12] or as conjugated linoleic acid (CLA) produced by several lactic acid bacteria that upregulate the expression of E-cadherin 1 and occludin in the gut [13]. Orally administered probiotics may have direct effects by stimulating immunological receptors, such as TLRs, on the epithelial and immune cells and/or can secrete metabolites, such as the short chain fatty acids (SCFA), with immunomodulatory effects. Furthermore, they can be taken up by M cells, processed and then presented to the DCs. In this way, probiotics modulate DC polarization and function, influencing the subsequent T and B cell responses at the local and extraintestinal levels [14]. The probiotic effects are more evident in the small intestine, where the number of endogenous bacteria is lower, but are also present in other body sites. 

In particular, probiotic bacteria can activate TLRs, leading to NF-kB and IRFs in immune cells. Lacticaseibacillus strains were able to prevent influenza A virus replication by activating type I IFN-dependent antiviral genes, and *L. paracasei* CNCM-I-1518 modified pro- and anti-inflammatory cytokine release in the lungs after influenza infection [15]. In mice, Bifidobacterium improved anti-influenza immune responses by decreasing IL-6 levels, also in the lungs, and higher IgG1 and IgG2 serum levels in probiotic-treated animals [16]. The LPS of Gram-negative probiotics is a strong inducer of IL-10 in PBMC that contributes to the induction of IgA at the mucosal level [17]. 

Regarding the mechanisms as adjuvant, Bron et al., reviewing the available data on the adjuvant activity of three strains of lactobacilli, according to the new taxonomic names of each strain/species of the Genus *Lactobacillus* [18], emphasized the role of lipoteichoic acid (LTA) of the *Lactiplantibacillus plantarum (Lp.plantarum)* strain, the secreted *Lacticaseibacillus rhamnosus* (*Lc. rhamnosus*) GG (also known as LGG) proteins p40 and p75, and the surface layer protein A (SlpA) of *Lactobacillus acidophilus (L. acidophilus)* NCFM [19]. These factors could represent the bacterial structural components supporting the vaccine adjuvant effect that has been shown since the pioneering papers from Link-Amster [20]. These authors enrolled a cohort of 30 adults fed with a fermented milk containing, in addition to yogurt bacterial cultures, one strain of *L. acidophilus* and one strain of *Bifidobacterium lactis,* which showed beneficial effects during a course of vaccination against *Salmonella thyphi* Ty21a. In particular, the treated group showed a 4.08-fold rise in specific IgA anti Ty2a LPS antibodies compared with a 2.48-fold rise in the control group [20]. After that combination, in vitro and animal models as well as human studies have provided further support to the potential role of probiotic bacteria as adjuvants. Since data from laboratory studies are promising and represent the biologically plausible rationale for their use, novel evidence underlines the ability of probiotics to strength the efficacy of vaccination.

Therefore, probiotics are considered biological response modifiers, and among these effects, the efficacy of adjuvants for boosting vaccine-specific immunity has been highlighted [21]. The effects of adjuvants on vaccination could be related to the increase in immunogenicity, the ability to influence the immunoglobulin response and the duration of immunity. 

For this reason, the role of probiotics as adjuvant factors to fortify the strategy regarding vaccinations has been investigated in this review of the literature. The data from in vitro and animal models as well as the results of our clinical investigations are presented.

## 2. In Vitro Studies

There is consensus that all Gram-positive bacteria may release LTA, wall teichoic acid, peptidoglycan (PG) and other wall components into the surrounding environment. All of these components are able to interact with the gut-associated lymphoid tissue (GALT) cells and with the systemic immune system of the subject [22]. These substances released by or produced through the metabolic activity of the microorganism may exert beneficial effects on the host, directly or indirectly, including an interaction with the immune system. They could also be present in non-viable bacteria, as has been suggested in the concept of post-biotics, if not destroyed or modified by the inactivation process. Furthermore, the role of the so-called “post-biotics” has been recently reviewed [23] and redefined as “preparation of inanimate microorganisms and/or their components that confers a health benefit on the host” [24].

In a closely related area of application, the use of probiotics as an adjuvant in anticancer therapy [25] showed that the probiotic strain LGG can selectively protect normal colon cells during radiotherapy protocols by acting as a “time-release capsule” that is able to gradually deliver the radio-protective LTA within the intestinal crypts. Therefore, it acts by selectively protecting the normal cells from radiation-induced cell death. The authors also noticed that LGG-derived LTA activates peri-cryptal macrophages, exerting a nonspecific, innate, immune system activation. The above-cited data may provide the molecular bases for the observed in vivo adjuvant effects discussed here below in animal models. 

## 3. Animal Vaccinations

Probiotic bacteria have been investigated in animal models both as adjuvants for vaccination in farm conditions (mainly for piglets and poultry) and as models to understand the adjuvant potential in human vaccinations. In this section, the focus is on the adjuvant role of probiotics in the vaccination of breeding animals. It seems noteworthy to point out that these trials have been performed in a substantial number of animal species, providing significant statistical potency. Moreover, interest in this particular use of probiotics has been raised in very recent years and mostly to support vaccination in the avian sector, which has been the source of epidemic attacks also against human beings.

As a matter of fact, very recently, it has been shown that a range of probiotic bacteria increases the efficacy of vaccinations in chickens treated with a vaccine specific for Newcastle disease [26] or avian influenza H9N2 [27,28]. In this case, adjuvant effects have been detected by using either bacteria cells [9] or spores [27]. In addition, turkeys were recently treated [29] with the herpesvirus of turkeys (HVT) vaccine to protect them against the emerging new strains of Marek’s disease virus (MDV). Bearing in mind the rapid selection of new viral variants that we experienced in these months, the results reported in this paper are of considerable interest. The authors tried to find a treatment that could protect against the emergence of new strains of MDV. Therefore, the study was carried out to investigate whether concurrent administration of probiotics with the HVT vaccine may enhance its protective efficacy against MDV infection. The results revealed that the administration of a probiotic to newly hatched chicks exerted a positive effect on host immune responses. This was evaluated as the percentage of tumors observed in the treated chickens in comparison with the controls. In the group treated with the vaccine alone, the percentage of tumors was 35.7%, approximately twice the percentage noted in the chickens treated with both the vaccine and probiotic lactobacilli (16.7%) [29]. Increasing interest in the veterinary sector regarding the use of probiotics as an adjuvant in breeding animal vaccination is also borne out by the fact that all of these papers were published in the last 18 months.

## 4. Animals as Model for Human Vaccinations

Pigs and piglets were mainly used as models to evaluate the adjuvant potential of probiotic lactobacilli, when vaccines against human viruses were used [30,31,32]. All of the papers clearly showed a beneficial immunological effect of the co-administration of vaccine and probiotics. The preferred animal/vaccine model was the piglets/rotavirus one, mainly because it was aimed at the pediatric use of this vaccine [33]. In the study by Vaslova and coworkers, a mixture of probiotic preparation was used, such as a combination of *Lc. rhamnosus* GG plus Bifidobacterium lactis BB12, and the piglets harbored conventional, piglet-specific intestinal microbiota [31].

The same probiotic combination was used by Chatta et al. in a different animal model: gnotobiotic piglets, i.e., germ-free piglets inoculated with human microbiota [18]. A similar animal model, gnotobiotic piglets with human intestinal microbiota, was then used in the studies carried out by Zhang [32], Wen [30,34] and Wang [35]. The probiotic used in all of these studies was LGG, and all the authors reported positive results, from the activation of regulatory signals during immune regulation to the positive effect of LGG as an adjuvant factor for attenuated human rotavirus [34,35]. LGG administration enhanced IFNγ+ T-cells and serum IgA antibody responses to the attenuated rotavirus. LGG at the higher dose was more effective at promoting stronger systemic IgG and IFN γ T-cell responses [34,35].

The conclusions drawn from the above-cited studies stated that the relationship between the modulation of gut microbiota and the regulation of host immunity by different doses of probiotics is indeed complex. Among the different probiotic strains, the LGG exerted divergent dose-dependent effects on the intestinal immune cell signaling pathway responses. However, the authors emphasized the need for further studies in order to gain further understanding of the dosage and the number of days of treatment. In other words, the results were promising but further research efforts are still needed. 

Animal models have also been used to assess the adjuvant efficacy of genetically modified probiotics [36]. The potential of genetically engineered probiotics has been reviewed, highlighting that metabolic engineering approaches enhance the probiotic properties as well as the production yield of these microorganisms. Consequently, the gene profiling and metabolic engineering approaches such as gene insertion, mutation, deletion or knock-out of the specific gene of these floras can be beneficial in modifying the organism to achieve enhanced probiotic properties [37].

## 5. Human Studies

Despite the favorable results from in vitro and animal models, the literature available on human studies is not univocal and is often highly heterogeneous. In a recent systematic review, Zimmermann et al. showed that a beneficial effect was reported in half of the studies, with oral vaccination and parental influenza vaccination exerting the greater effect on vaccine response [38].

The data regarding probiotic supplementation on influenza vaccine seroconversion in adults have been reported in a recent meta-analysis [39]. Investigating the effect of probiotics on immune response to influenza vaccination in adults, the authors found a significant improvement in the H1N1, H3N2 and B strain serum-protection rate, which suggests that probiotics are effective in increasing immunogenicity by influencing seroconversion in adults vaccinated for influenza [39]. Elderly subjects who were given a yogurt drink containing *L. casei* had higher influenza-specific antibody titers [40], and in healthy adults, *Limosilactobacillus fermentum (Ls. fermentum)* improved influenza vaccine immunogenicity [41]. In another study, *Lc. rhamnosus* LGG administration ensured an increased protective rate for one strain but not for all strains present in that influenza vaccine in comparison to placebo [42]. In a recent paper by Bianchini and colleagues, the effect of LGG administered for 3 months was tested on immune responses to an influenza vaccine in children and adolescents with type 1 diabetes. The results showed that the influenza vaccine was highly effective and safe in this population and that LGG administration did not modify the vaccine humoral responses versus the placebo-treated group. The LGG supplemented group presented reduced inflammatory responses from activated peripheral blood mononuclear cells (PBMCs), underlying the possible anti-inflammatory effects given by the systematic administration of the probiotics [43].

Data regarding infants and children are more heterogeneous. Isolauri et al. [44] demonstrated a significant effect of LGG on rotavirus IgA and IgM seroconversion after vaccination in infants of 2–5 months of age. The LGG supplementation was administered immediately before and for 5 days after the vaccine. In another study, Kukkonen et al. evaluated the effect of a mixture of four probiotics, given to the mothers from the 36th week and then to the newborns during the first 6 months of life [45]. The infants were vaccinated with diphtheria, tetanus and pertussis at 3, 4 and 5 months and with anti-Hemophilus B (HiB) at 4 months. Antibody responses were evaluated at 6 months, showing a significantly increased seroconversion rate in response to vaccination for HiB but not for tetanus and diphtheria in the probiotic-treated group. For the latter, however, the probiotic supplementation did not negatively affect the vaccination outcomes, since the IgG titers were comparable between groups [45].

In another study, Taylor and colleagues administered *Lc.* LAVR1-A1 daily for the first 6 months of life. Then, the infant PBMCs were isolated and stimulated with various factors in vitro [46]. The authors demonstrated that the administration of probiotics was accompanied by a reduced production of cytokines after polyclonal stimulation but found no significant effect of immunomodulation to both Th1 and Th2 responses to allergens or other stimuli. Even though no direct investigation on the effect on the vaccine response was determined, they concluded that probiotics may have immunomodulatory effects on vaccine responses [46]. In a different study, in which infants received a cereal supplemented with *Lc. paracasei* F19 from 4 to 13 months of life, there was an increased response to the diphtheria vaccine, with no effect on other types of vaccines administered to the infants [47]. Soh et al. demonstrated a positive influence on the anti-HBsAg responses in infants supplemented for the first 6 months of life, again, with a mixture of probiotics, *Lc. rhamnosus* LPR and *Bifidobacterium longum*. However, this trend was not statistically significant, and no effect was observed in a modified vaccination schedule [48].

Finally, an Australian group utilized the LGG strain only in the pre-natal period, administering supplementation to mothers from 36 weeks to delivery in comparison with a placebo. No variation in the diversity of the microbiota of the offspring [49] and in the incidence of eczema [50] was observed. Furthermore, in the same study group, Licciardi found that supplementation during pregnancy, starting from 36 weeks of gestational age with LGG, produced a significant reduction in the immunological response to tetanus, HiB and pneumococcus vaccines. The authors also reported an increase in regulatory T cells. This was a high-risk allergy cohort, and effectively, more infants in the group of mothers treated with LGG developed eczema or atopy [51]. The authors speculate that the response to vaccines may have been influenced more by the atopic status of the offspring rather than by the effect of LGG supplementation. In fact, in the atopy status, the predominantly Th2-biased response could downregulate the Th1-based IgG levels that usually follow vaccination [51].

## 6. Conclusions

In conclusion, the use of probiotics as adjuvant factors in vaccination is a subject of great interest but is still under debate. It could represent a strategic argument for the application of vaccines, both in the veterinary as well in the medical fields. The biological plausibility of the adjuvant effect, at least for some specific bacterial strains, is supported by clear demonstrations from both in vitro and in animal models. A recent review [52] listed the potential mechanisms supporting the adjuvant action of probiotics, which modifies the microbiota composition. The authors suggested that microbe-associated molecular patterns (MAMP) and metabolites such as SCFA are the most probable candidates as promoters of the adjuvant action.

The huge differences observed between studies in the timing of intervention with probiotic supplementation (when and for how long), characteristics and age of the patients (pregnant women, newborn infants, adults, the elderly, healthy or high-risk subjects), the probiotic strain utilized and the type of vaccine (live attenuated, killed/inactivated and subunit/recombinant) greatly influence the conclusions. 

We believe that further research studies are mandatory in this field, as the in vitro data so far available provide a solid rationale of use and yield some clues about the mechanisms of action, strongly suggesting that the use of probiotics as adjuvants should be considered in vaccination. Probiotics provide a relatively inexpensive mode of intervention to improve vaccine efficacy and the duration of protection [38].

Specific attention has to be given to the strain used, since the efficacy of action is absolutely strain-related. Lactobacilli and LGG, in particular, are the most studied and promising among the probiotics widely utilized to date for these aims. 

In this COVID-19 pandemic, nutritional supplements such as probiotics with antimicrobial and immunomodulatory activities are promising therapeutic adjuvants for the treatment of COVID-19 and for the prevention of viral spread [53]. The present COVID-19 pandemic situation underlines how all efforts that can ameliorate vaccine efficacy should be attempted and can be a stimulus to promote further research. Since different vaccine platforms are used in COVID-19 vaccine development, we believe that the use of probiotics in this development should be considered. Indeed, probiotic supplementation as adjuvants in boosting immunity and in enhancing vaccine-specific responses could be important in the general population and, in particular, in the elderly and in children, where the effectiveness and duration of immunization could have even more bearing.

## Figures and Tables

**Figure 1 vaccines-09-00811-f001:**
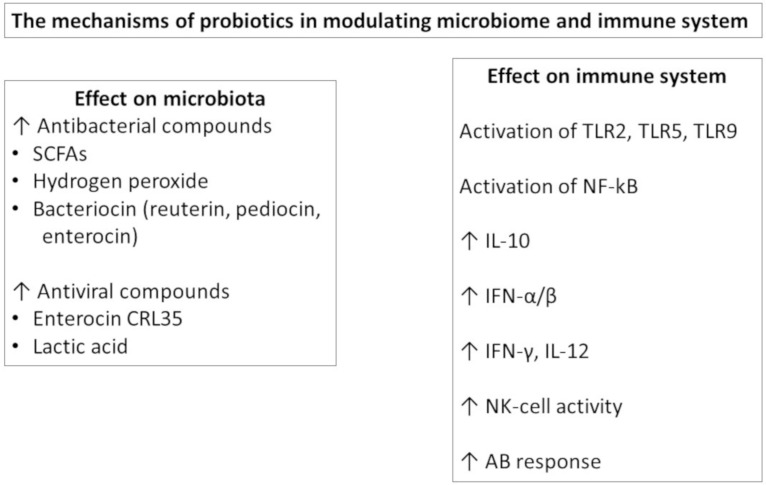
The mechanisms of probiotics in modulating.

## Data Availability

Not applicable.

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
