# Peer review of "Probiotics as Adjuvants in Vaccine Strategy: Is There More Room for Improvement?"

_vaccines, 2021, doi:10.3390/vaccines9080811_

Round 1

Reviewer 1 Report

The authors Peroni et al. have critically reviewed a wide breadth of literature the most emerging topic on use of probiotics as adjuvant therapy with vaccines to improve the immune system. 

A few comments:

  1. Although the authors extrapolated the role of probiotics altering microbiome thereby improving immune system. However, the overall topic needs more explanation. Authors can provide a possible mechanistic illustration.
  2. Can authors provide more information, on dosing, type of probiotics, what is the role of genetically modified probiotics with vaccine?
  3. Vaccine-induced antibodies/immune cells interaction with probiotic-induced immune responses. 
  4. Different vaccine platforms are used in COVID-19 vaccine development, what is scope of probiotics in these conditions.

Author Response

Thank you very much for your comments. We are grateful to your considerations that allow us to improve the manuscript.

  1. We better explained the role and mechanisms of probiotics in altering microbiome and in turn modifying the immune system. We inserted a new paragraph titled “Microbiome, probiotics, immune system and vaccines”, where all these aspects have been reviewed and explained (from line 34 to line 78). A figure on mechanistic processes has also been created according to suggestion.
  2. We have inserted a reference (ref 37) and sentences (lines 180-186) on this aspect that is very important for the future research.
  3. This topic has been also explained in more details along the ms.
  4. Regarding the probiotic use in COVID-19 vaccine development we believe that this could be very interesting topic for the next future. We have inserted this concept in the discussion section (line 281).

Reviewer 2 Report

In this review, authors investigate the current researches of probiotics as adjuvants in vaccine strategy. The manuscript is well organized, described briefly but comprehensively including laboratory experiments, in vitro studies, in vivo studies in animals and clinical studies in human. Moreover, the authors analyzed the potential reasons for heterogeneous effects in human studies, and encouraged more probiotics and clinical studies in developing probiotics as adjuvants to improve vaccines in combatting pathogens. This manuscript will provide a good reference for applying probiotics as adjuvants to enhance vaccine therapy especially during the current virus crisis period.

Author Response

Reviewer 2.

We appreciated the positive comments of the reviewer and we thank for the positive evaluation.

Reviewer 3 Report

Although the literature is full of probiotics research papers, some with sloppy, superficial experiments, this review by Peroni and Morelli breathes fresh air into this stale area.  In this short review authors have taken a straightforward, unbiased approach of presenting both the pluses and minuses of the main types of probiotics tested and reported. While acknowledging a lack of consensus in human applications, mainly due to small number of subject pool and diverse conditions used by different researchers, the authors also presents this as reason for the need of more controlled studies, and hence there is room (“space”?) for further improvement. This is based on several successful use of probiotics as adjuvants in other systems.

I have a few minor comments and suggestions, as follows:

(1) “Revision of literature” is probably fine, but a more common expression would be “review of literature”.

(2) It is true that adjuvant field has remained an arbitrary science. Nevertheless, in this review, the authors could present mechanistic aspects of the probiotics adjuvant effect where possible. Specifically, they can state whether some of the bacterial constituents act as PAMPs and activate the PRRs in the innate immunity pathway to promote better overall immune response. Various forms of bacterial DNA, LPS, and flagellin, for example, can activate the TLR pathways, in turn activating dendritic cells, etc.

(3) Similarly, no statement is made about how HVT may act as “adjuvant” against MDV and its variants (Line 94 area).

(4) In several places, the authors appear to refer to the current Covid-19 pandemic simply as “pandemic” without mentioning Covid-19 (Line 203, 218).  This should be corrected.

(5) “Limosilactobacillus. Fermentum” (Line 145) should be correctly written as “Limosilactobacillus fermentum”, the species name starting with lowercase letter.

Author Response

We thank the reviewer for the very positive comments. The title has been changed accordingly to suggestion to “more room for improvement”.

  1. We changed to “review” of literature along the paper.
  2. The mechanistic aspects of the probiotic adjuvant effects have been deeply clarified along the paper and we have created a Fig that synthetize the different aspects.
  3. The pandemic was always associated to the term COVID-19 as suggested.
  4. The term was corrected accordingly.

Reviewer 4 Report

This article focuses on a very relevant topic with high interest. The abstract suggests the information gathered spans across laboratory investigations and clinical studies, however the research cited in this paper is somehow limited.

The article is defined as “a revision of the literature” however the “original” which this “revision” was based on was not mentioned. There was already good review on this topic (such as the one by Zimmermann and Curtis, published in Vaccine, Volume 36, Issue 2, 4 January 2018, Pages 207-213) where many aspects mentioned in this current article have been discussed in great detail.

In my opinion, this article can be largely improved if the author can include more meta analyses on this topic, and provide own views on both the promising and shortcoming aspects in this research field. Focusing on topics that weren’t covered extensively in existing reviews, such as the use of animal models and human trials in different age groups, will greatly supplement the current summary of knowledge.

Author Response

We are indebted to the reviewer for the useful comments. As suggested, we have inserted more meta-analyses on this topic, increasing the number of references in the actual version, in particular ref. 3, 11, 14, 38, 52. We have inserted accordingly more references dealing with the topic to better explain the knowledge on the topic. The conclusions of the review by Zimmermann have been inserted (line 188-191). This was very useful to provide a better understanding of the existing data and analyses on the topic giving consequently the chance to consider the promising and shortcoming aspect in this research field. we also provided our own views on this topic for the very next future.

Round 2

Reviewer 4 Report

The authors have addressed to the comments accordingly. The revision enables the provision of a more holistic picture on this topic. I have only a few minor suggestions:

Line 73: "L. paracase" should be in italic.

Line 266: suggest revise the word "cheap" to "inexpensive".

Line 277: "a probiotic use" to "the use of probiotic".

Author Response

We thank the reviewer for the comments that improved our ms.

All the suggested points have been inserted in the text marked in green in the marked version.